# The Exon-Based Transcriptomic Analysis of Parkinson’s Disease

**DOI:** 10.3390/biom15030440

**Published:** 2025-03-19

**Authors:** Sulev Kõks

**Affiliations:** 1Personalised Medicine Centre, Health Futures Institute, Murdoch University, Perth, WA 6150, Australia; sulev.koks@murdoch.edu.au; Tel.: +61-(0)-8-6457-0313; 2Perron Institute for Neurological and Translational Science, Perth, WA 6009, Australia

**Keywords:** Parkinson’s disease, transcriptome, whole transcriptome analysis, exons, RNA-seq, PPMI, blood transcriptome

## Abstract

Parkinson’s disease (PD) is a neurodegenerative disease with a complicated pathophysiology and diagnostics. Blood-based whole transcriptome analysis of the longitudinal PPMI cohort was performed with a focus on the change in the expression of exons to find potential RNA-based biomarkers. At the moment of diagnosis, the expression of exons was very similar in both control and PD patients. The exon-based analysis identified 27 differentially expressed exons in PD patients three years after the diagnosis compared to the health controls. Moreover, thirteen exons were differentially expressed during the three-year progression of the PD. At the same time, control subjects had only minimal changes that can mostly be attributed to being related to aging. Differentially regulated exons we identified in the PD cohort were mostly related to different aspects of the pathophysiology of PD, such as an innate immune response or lysosomal activity. We also observed a decline in the expression of the *OPN1MW3* gene that is related to colour vision, which suggests that colour vision analysis could be a practical biomarker to monitor the progression of PD.

## 1. Introduction

Gene expression analysis can be focused on a single gene and a selected group of genes, or it can include all genes in the genome [1]. Whole transcriptome, or genome-wide gene expression profiling, is a very promising tool and gives a good snapshot of the biological system or describes the response to the stimuli [2,3,4]. This analysis can be used for any kind of biosample and even for environmental samples as metatranscriptome analysis [5,6]. Understanding the alterations in the RNA expression caused directly by the disease-causing mutation or by the interaction of the mutated and dysfunctional gene product is useful [7,8]. Moreover, RNA profiling can be very useful in describing the molecular changes related to pathological conditions, and this way they can help to identify potential RNA-based biomarkers [9,10].

Parkinson’s Disease (PD) is one of the most prevalent neurological diseases, characterised by several well-established causative genetic replacements and many associated genetic variants [11,12,13,14]. The pathological details of PD are still misunderstood, and, in providing the complex interactions of the neuropathology of PD, the genomic mechanisms causing the pathology are similarly likely to be both multifactorial and complex [15,16,17]. The transcriptomic analyses have been attempted in several studies to identify the signatures of PD and, as a result, have identified transcriptional signatures or specific splicing events of genes expressed in the basal ganglia [18,19]. However, considering the issues involved in obtaining brain samples from patients, peripheral tissue biomarkers are in high demand. The analysis of transcriptomes from the easily accessible peripheral tissues (skin, keratinocytes, and blood) in the context of CNS expression data of PD patients has identified remarkably significant differences between PD and control transcriptomic profiles [15,20,21]. These studies underline the significance of the analysis of the peripheral tissue not only to determine biomarkers that can predict the presence or absence of the disease but also to gain knowledge and improve the understanding of neurodegenerative diseases. In this study with a longitudinal design, we analysed the blood exonic transcriptional changes of PD patients and healthy controls (Table 1) collected at two different time points of disease progression, either at the time of diagnosis (baseline, BL) or three years later as a follow-up (V08). The general correlation between blood and brain gene expression is limited but informative, with both tissues having distinct gene expression profiles. The blood lacks neurons, and many brain-specific cell types, and the overall correlation is low to moderate (Pearson’s r = 0.5). However, some gene pathways, especially those related to inflammation, immune function, and metabolism, are active in both tissues and show a stronger correlation. Some previous studies suggest that blood-based RNA biomarkers could be helpful in the early detection of Parkinson’s disease and other brain disorders. However, validation and specificity remain challenges. Our study, using a large longitudinal dataset, aimed to provide additional evidence of the suitability of blood RNA as a biomarker for PD.

Blood-derived whole transcriptome data was a starting point for our analysis, but instead of counting the reads mapping to genes, we evaluated the reads that map the specific exons. This reads mapping to the exonic sequences gives higher confidence to the short-read sequencing to detect specific transcript isoforms. We utilised whole transcriptome blood data from the Parkinson’s Progression Markers Initiative (PPMI) cohort, which includes information on both Parkinson’s disease (PD) subjects and healthy control (CO) individuals at two different stages in the progression of PD.

## 2. Materials and Methods

### 2.1. Datasets

In this research, we employed data from the Parkinson’s Progression Markers Initiative (PPMI) cohort, which was retrieved from www.ppmi-info.org/data, accessed on 16 May 2021. The PPMI serves as a longitudinal cohort for individuals with Parkinson’s disease, aiming to characterise the progression and identify biomarkers associated with the condition. The dataset includes whole transcriptome information from blood samples, along with genetic and clinical data. For the RNA sequencing, 1 μg of RNA extracted from PaxGene tubes was utilised, and the sequencing was conducted at Hudson’s Alpha’s Genomic Services Lab using an Illumina NovaSeq6000 (Illumina, Inc., San Diego, CA, USA). Each sample underwent rRNA and globin reduction before directional cDNA synthesis was carried out using the NEB kit. After the second-strand synthesis, the library samples were prepared utilising the NEB/Kapa (NEBKAP) library kit. Fastq files were combined and aligned to GRCh38p12 using STAR (v2.6.1d) with reference to GENCODE v38.

In the current study, we analysed the blood whole transcriptome data of 1074 subjects (Table 1), 346 healthy controls and 728 PD samples, to find differentially expressed exons.

### 2.2. Workflow

Bam files were loaded into the R environment (version 4.4.2), and RNA reads corresponding to exons were identified using the GenomicFeatures (version 1.58.0) and GenomicAlignments (1.42.0) packages. Differential expression analysis was conducted utilising DESeq2 (1.46.0), and exons corrected for False Discovery Rate (FDR) were identified and reported, followed by further pairwise comparisons.

### 2.3. Statistical Analysis

A formal statistical analysis to assess the differential expression of the exons was conducted utilising the “DESeq2” package in R, with only those exons with a False Discovery Rate (FDR) below 0.1 being reported. Additionally, selecting FDR-adjusted exons was employed for pairwise comparisons of the expression data using the Wilcoxon test, and the resulting plots were created with the “ggpubr” (version 0.6.0) package.

## 3. Results

### 3.1. General Outcome (Table 2)

We examined the exonic expression patterns in PD and CO individuals at the time of diagnosis (BL visit) and three years afterwards (V08). By using gencode v38 for annotation, we detected 834,912 exons in all samples. In all our comparisons at baseline between the PD and CO groups, there was minimal variation in exon expression (Table 2). After three years of diagnosis of PD, 27 exons were differentially expressed. In addition, only nine differentially expressed exons in CO subjects were observed if we compared them at BL and V08 time points (Table 2). In the case of PD patients, thirteen exons were differentially expressed (Table 2) in longitudinal analysis.

### 3.2. Comparisons Between PD and CO and Between Visits (BL Versus V08)

#### 3.2.1. At the Time of PD Diagnosis, Baseline (BL)

At the time of the first visit and the moment of PD diagnosis, only two exons were differentially expressed (FDR < 0.1) between controls and PD patients, ENSE00003719556 and ENSE00003746162 (Table 3). These exons are from the genes RP11-403I13.4 and RP11-596C23.2, respectively. These are both long non-coding RNAs (lncRNAs); RP11-403I13.4 (alias lnc-PPIAL4F-4-5) has seven exons with exon 5 upregulated, and RP11-596C23.2 (alias lnc-ARF6-5) has 1 exon that was upregulated. RP11-403I13.4 is also known as a novel pseudogene, similar to phosphodiesterase 4D interacting protein PDE4DIP.

#### 3.2.2. Three Years After the PD Diagnosis Compared to Controls, V08

Twenty-seven exons were differentially expressed in PD patients compared to control subjects (Table 4, Figure 1) after three years from the start of the study. Six exons were upregulated from five genes TRDC, TRDV2, MAN1A2, DDI2, and AE000661.37. TRDC is a T-cell receptor delta constant gene, TRDV2 is a T-cell receptor delta variable 2 gene, MAN1A2 is a mannosidase alpha class 1A member 2, DDI2 is DNA-damage inducible 1 homolog 2, and AE000661.37 is a lncRNA gene.

The rest of the exons were downregulated, with exon 5 for RP11-403I13.4 and exon 2 MST1P2 (macrophage stimulating 1 pseudogene 2) most significantly downregulated. In addition, we found FAM20C (FAM20C Golgi-associated secretory pathway kinase) and RAB11FIP3 (RAB11 family interacting protein 3) significantly downregulated in PD patients at the V08 timepoint. Exon seven from the SCAF1 (supercomplex assembly factor 1), exon 2 of the NRGN (neurogranin) gene, and exon 2 of the G0S2 (G0/G1 switch gene 2) were also significantly downregulated.

#### 3.2.3. Longitudinal Change After the PD Diagnosis Compared to Baseline in the PD Group

Three years after receiving a diagnosis, thirteen exons were differentially expressed in PD patients compared to the start of the disease (Table 5, Figure 2). Some notable findings were the following. Exon 11 of the MST1P2 was downregulated, and exon 2 of MAN1A2 was upregulated. These are the exons that were also differentially expressed in the PD–CO comparison at V08.

Exon 1 for RP11-403I13.4 and exon 2 CAPN6 (calpain 6) had the lowest log2FC values and were the most downregulated exons in PD patients three years after the start of the disease. Exon 6 of OPN1MW3 (opsin 1, medium-wave-sensitive 3) and exon 2 of GDAP1 (ganglioside-induced differentiation-associated protein 1) were downregulated in PD patients compared to their initial visit.

#### 3.2.4. Longitudinal Change After Three Years Compared to Baseline in the CO Group

This comparison shows age-related changes in healthy people. Three years after the start of the study, we identified nine exons to be differentially expressed in healthy controls (Table 6). All these exons were upregulated. Exon 5 of the gene RP11-403I13.4 and exon 2 from the gene MST1P2 were the most upregulated exons. Exon 1 from the IGHV3-21 (immunoglobulin heavy variable 3-21) gene and exon 1 of the gene RNUY1 (RNA, Ro60-associated Y1) were upregulated, too. The rest of the upregulated exons were from small nucleolar RNA genes.

### 3.3. Pairwise Comparisons for Differential Exon Expression

#### 3.3.1. All Samples Combined Longitudinal Change

To obtain a more general understanding of the time-related or ageing-related changes in exons, we started with a combined sample (PD and CO together) and performed a pairwise comparison of all the exons that showed differential expression in the previous section. We detected only six exons to be differentially expressed with statistical significance. These exons were exon 5 of the RP11-403I13.4-002, exon 2 of the MAN1A2-201, exon 1 of the SNORA13-201, exon 1 of the IGHV3-21-201, exon 1 of the RNVU1-2-201, and exon 2 of the RP11-144F15.1-001 transcripts (Figure 3). Exon 5 of the RP11-403I13.4-002 and exon 2 of MAN1A2-201 were upregulated, exon 1 of the SNORA13-201, exon 1 of the IGHV3-21-201, and exon 1 of the RNVU1-2-201 were upregulated by ageing, while exon 2 of the RP11-144F15.1-001 was downregulated after three years from the beginning of the study. These exons are considered to be related to ageing.

#### 3.3.2. Change in PD Patients Three Years After Diagnosis

We next analysed only patients with PD and performed pairwise testing for the thirteen exons we identified before in whole transcriptome analysis (Table 5). Out of all thirteen exons, only three exons were statistically significant in pairwise comparison (BL versus V08, Figure 4). Exon 2 of the MAN1A2-201 transcript was upregulated at visit V08. Exon 2 of the RP11-144F15.1-001 and exon 6 of the OPN1MW3-201 transcripts were downregulated in PD patients after three years from the diagnosis.

#### 3.3.3. PD Progression-Related Changes Three Years After Diagnosis

We next analysed only data from the visit V08, three years after the start of the study, and we compared CO to the PD patients. This way, we could detect the exons that are expressed specifically for the progression of the PD disease. We performed pairwise testing for the twenty-seven exons we identified before in whole transcriptome analysis (Table 4). Out of all initially detected exons, nine exons were statistically significantly expressed in pairwise analysis (CO versus PD, Figure 5). Exon 2 of the MAN1A2-201, exon 1 of the TRDC-201, exon 4 of the TRDC-201, exon 3 of the AE000661.37-007, exon 5 of the DDI2-202, and exon 2 of the TRDV2-201 transcripts were all upregulated in PD patients after three years of the disease progression. Exon 1 of the FAM20C-201, exon 1 of the CABLES2-201, and exon 9 of the SH2B2-202 transcripts were all downregulated in PD patients during the progression of the disease.

## 4. Discussion

In the present study, we used the PPMI dataset of PD patients and healthy controls, and we compared the exonic expression between the baseline and three years after the diagnosis. The study is based on the blood whole transcriptome data, and we focused on the differential expression exons. We used gencode v38 for the annotation of whole transcriptome data, and we detected 834,912 exons in all samples. As a result, we detected PD-specific exons and progression-specific exons.

At the start of the study, at the moment of diagnosis, PD patients and healthy controls had only two exons differentially expressed, RP11-403I13.4 and RP11-596C23.2 (alias lnc-ARF6-5). This shows that early stages of the disease blood transcriptomics are not severely affected in PD patients. RP11-403I13.4 is a novel pseudogene; it is similar to phosphodiesterase 4D interacting protein PDE4DIP. RP11-403I13.4 has been found differentially expressed in our previous study, where we analysed intronic expression in the PPMI cohort [22]. Phosphodiesterase 4D interacting protein (PDE4DIP) or myomegalin is a protein that serves to anchor phosphodiesterase to the Golgi in the cell [23]. A study using the SWIM algorithm to identify genes associated with dementia found that PDE4DIP is a key factor in the development of Alzheimer’s disease, vascular dementia, and frontotemporal dementia [24]. However, the functions of this new pseudogene and RP11-596C23.2 are not known.

Three years on, a comparison between PD and CO reveals significant differences in the expression of exons. Most interestingly, one gene that is upregulated is MAN1A2, which is a gene for mannosidase alpha, class 1A member 2. This gene has been recognised as an RNA-based blood biomarker for early PD in many independent studies [15,25,26,27]. Alpha-mannosidase is a lysosomal enzyme that may be a feasible biomarker for PD, and the activity of alpha-mannosidase in CSF may reflect pathological changes in the brain associated with neurodegenerative disorders like PD. In one study, the activity of alpha-mannosidase was reduced by 24–71% in the CSF of PD patients [28]. In our study, exon two from the specific transcript MAN1A2-201 was elevated in PD patients at visit V08 compared to the controls. Moreover, the same exon 2 of the MAN1A2-201 transcript was also significantly elevated in PD patients after three years of progression of the disease (BL versus V08 in the PD group only). Therefore, MAN1A2 seems to be a very good biomarker for cross-sectional design, and it also reflects the progression of the disease.

The other interesting cluster of genes that were upregulated in PD patients three years after the diagnosis were related to T-cell receptors: exon 1 and exon 4 from the TRDC-201 (T cell receptor delta constant), exon 2 from the TRDV2 (T cell receptor delta variable 2), and exon 3 from the AE00061.37 that is a transcript TRD-AS1-207 of the gene TRD-AS1 (T cell receptor delta locus antisense 1). All these exons were significantly upregulated in PD patients compared to the controls and, therefore, are potentially very good biomarkers for PD. Some previous studies have identified the increased level of delta T-cells in PD patients, and the role of T-cells in PD pathogenesis has been suggested [29,30,31].

Our analysis also identified downregulation of exon 2 of G0S2-201, exon 7 of SCAF1-201, and exon 2 of NRGN-201 transcripts. These are all genes that have been linked to PD in previous studies, but their precise involvement and their biomarker value are still uncertain [15,32,33]. NRGN, or neurogranin, is a gene that is involved in PD and is a well-known potential biomarker. SCAF1 regulates mitochondrial function. Therefore, these are well-justified potential biomarker candidates from exon expression profiling.

The longitudinal analysis of the gene expression profiles in PD patients gave us several significantly changed genes, with OPN1MW3 as one of the most promising. OPN1MW3 is a protein-coding gene for the cone pigment opsin 1, specifically for the medium-wave-sensitive opsin 3. In the present study, we detected significant downregulation of the exon 6 of the OPN1MW3 gene in PD patients after three years of diagnosis. Interestingly, reduced colour vision has been described in PD patients, and it is considered to be a specific feature of PD compared to Alzheimer’s Disease [34,35]. Melanopsin-mediated pupil function is impaired in PD [36]. In addition, the loss of DJ-1 has been shown to elicit retinal abnormalities and visual dysfunction [37]. This finding is in very good concordance with the data showing the decline of melanopsin cells in the retina of PD patients [38,39]. The degeneration of the retina can also explain the disordered sleep and circadian rhythms in PD patients. This can also suggest that the expression of the OPN1MW3 gene could be used as a progression biomarker in combination with the testing for the changes in colour vision [40]. Therefore, the decline of colour vision could be a potential biomarker for PD progression, and it fits very well with the blood transcriptome data.

## 5. Conclusions

This study performed exon-based whole transcriptome analysis in a longitudinal PD cohort, PPMI. We identified a large number of exons and transcripts specifically expressed in PD patients with very few possibly age-related changes in the control group. The exon-based analysis helped to identify specific transcripts that were differentially expressed in PD patients and were related to the disease progression. Some of the transcripts were suitable for the cross-sectional analysis and separated PD from controls.

## Figures and Tables

**Figure 1 biomolecules-15-00440-f001:**
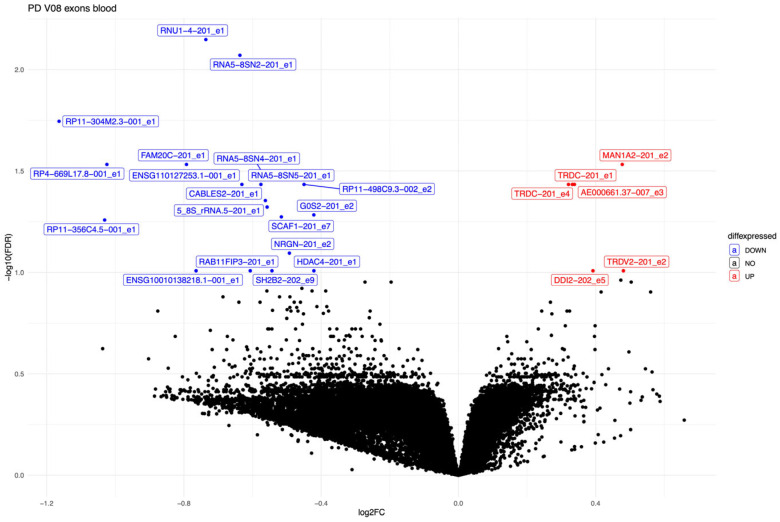
Volcano plot of the differentially expressed exons at the visit V08 in PD patients compared to the controls.

**Figure 2 biomolecules-15-00440-f002:**
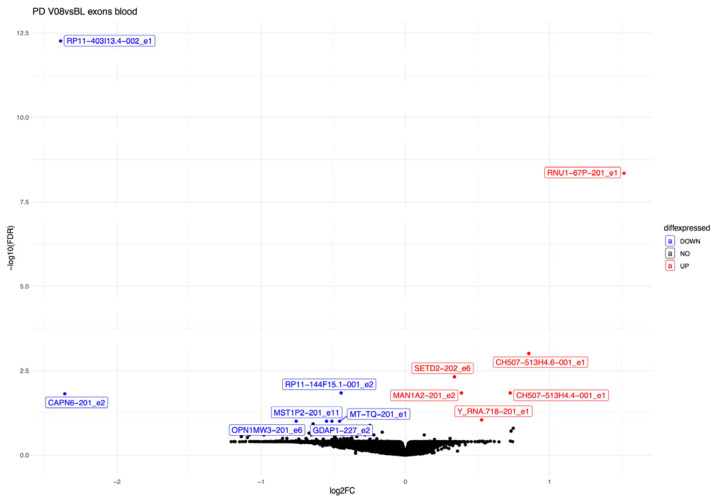
Volcano plot of the differentially expressed exons in PD patients at visit V08 compared to the baseline visit.

**Figure 3 biomolecules-15-00440-f003:**
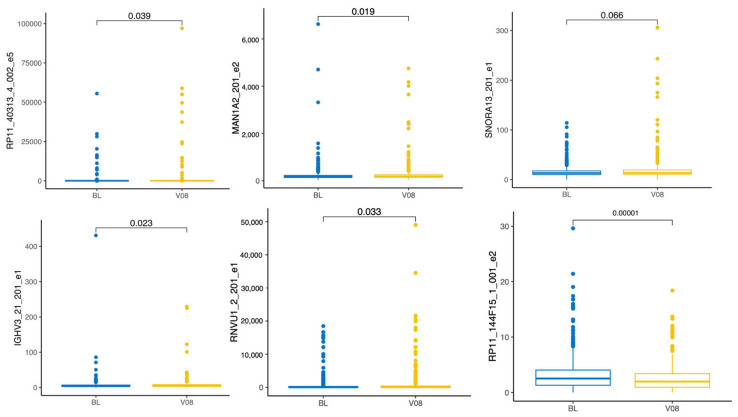
Boxplot of the differentially expressed exons in combination with PD and CO subjects within three years. The comparison is between the baseline (BL) and visit V08. Statistical significance was tested with the Wilcoxon test.

**Figure 4 biomolecules-15-00440-f004:**
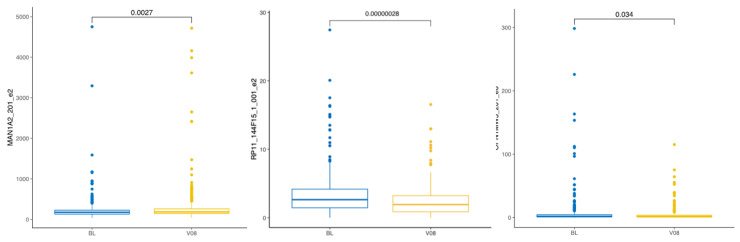
Boxplot of the differentially expressed exons in patients with PD within three years. The comparison is between the baseline (BL) and visit V08. Statistical significance was tested with the Wilcoxon test.

**Figure 5 biomolecules-15-00440-f005:**
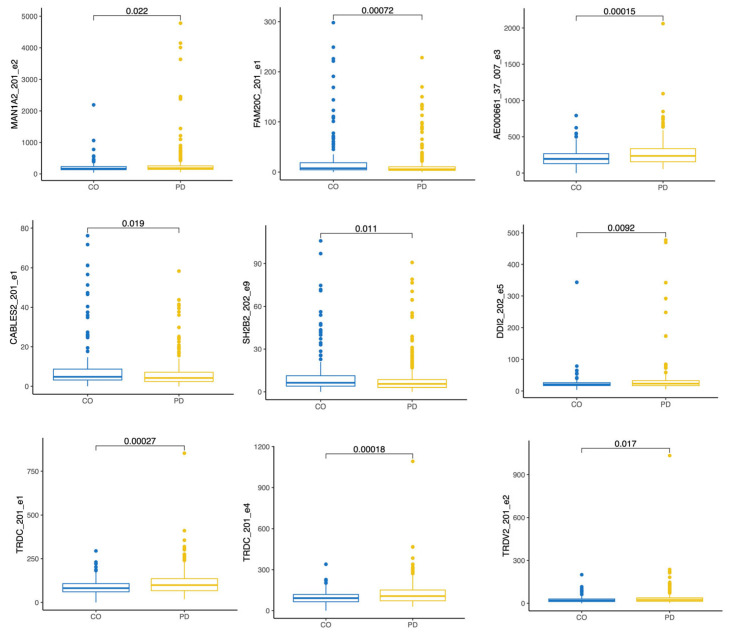
Boxplot of the differentially expressed exons in study subject after three years, at visit V08. The comparison is between the controls (CO) and Parkinson’s patients (PD) at visit V08. Statistical significance was tested with the Wilcoxon test.

**Table 1 biomolecules-15-00440-t001:** Overview of the study samples and design. Subject numbers with the blood whole transcriptome data of the PPMI cohort are given.

Study Group	Baseline (BL)	Three Years (V08)
CO	189	157
PD	390	338

**Table 2 biomolecules-15-00440-t002:** Analysis of the differential expression of exonic reads. The baseline (BL) was compared to the three-year (V08) time points between the PD and CO groups. The impact of time (three years) was evaluated for the PD and CO groups (V08 compared to BL), with a significance threshold set at FDR < 0.1.

Group	Differentially Expressed Exons	Comparison
BL	2	PD versus CO
V08	27	PD versus CO
PD	13	V08 versus BL
CO	9	V08 versus BL

**Table 3 biomolecules-15-00440-t003:** Differentially expressed exons and transcripts at first visit, at the start of PD compared to healthy controls. Two exons (e) had a False Discovery Rate (FDR) below 0.1.

Exon ID	Log_2_ FC	FDR	Transcript and Exon
ENSE00003719556	1.6	1.8 × 10^−7^	RP11-403I13.4-002 e5
ENSE00003746162	0.5	0.06	RP11-596C23.2-001 e1

**Table 4 biomolecules-15-00440-t004:** Differentially expressed exons and transcripts at the last visit, three years after diagnosis of PD, compared to healthy controls (CO). Twenty-seven exons (e) had a False Discovery Rate (FDR) below 0.1.

Exon ID	Log_2_ FC	FDR	Transcript and Exon
ENSE00003719556	−2.8	1.03 × 10^−21^	RP11-403I13.4-002 e5
ENSE00001331364	−2.0	1.55 × 10^−16^	MST1P2-201 e2
ENSE00001806909	−0.7	0.007	RNU1-4-201 e1
ENSE00003742227	−0.6	0.008	RNA5-8SN2-201 e1
ENSE00002197845	−1.1	0.017	RP11-304M2.3-001 e1
ENSE00001635177	0.5	0.029	MAN1A2-201 e2
ENSE00002058739	−1.0	0.029	RP4-669L17.8-001 e1
ENSE00001232590	−0.8	0.029	FAM20C-201 e1
ENSE00003725298	−0.6	0.032	RNA5-8SN4-201 e1
ENSE00001885147	0.3	0.037	TRDC-201 e1
ENSE00002496627	0.3	0.037	TRDC-201 e4
ENSE00002533311	0.3	0.037	AE000661.37-007 e3
ENSE00002663611	−0.4	0.037	RP11-498C9.3-002 e2
ENSE00002724720	−0.6	0.037	ENSG110127253.1-001 e1
ENSE00003734991	−0.6	0.037	RNA5-8SN5-201 e1
ENSE00001820608	−0.6	0.044	CABLES2-201 e1
ENSE00003746458	−0.6	0.048	5_8S_rRNA.5-201 e1
ENSE00001443289	−0.4	0.052	G0S2-201 e2
ENSE00001387755	−0.5	0.053	SCAF1-201 e7
ENSE00002592795	−1.0	0.055	RP11-356C4.5-001 e1
ENSE00001014304	−0.5	0.080	NRGN-201 e2
ENSE00001351972	0.4	0.098	DDI2-202 e5
ENSE00001242645	−0.4	0.098	HDAC4-201 e1
ENSE00002480538	−0.5	0.098	SH2B2-202 e9
ENSE00001508095	0.5	0.098	TRDV2-201 e2
ENSE00002222336	−0.6	0.098	RAB11FIP3-201 e1
ENSE00003752463	−0.8	0.098	ENSG10010138218.1-001 e1

**Table 5 biomolecules-15-00440-t005:** Differentially expressed exons and transcripts in PD patients three years after diagnosis of PD, changes from the BL. Thirteen exons (e) had a False Discovery Rate (FDR) below 0.1.

Exon ID	Log_2_ FC	FDR	Transcript and Exon
ENSE00001740173	−2.4	5.5 × 10^−13^	RP11-403I13.4-002 e1
ENSE00001806816	1.5	4.5 × 10^−9^	RNU1-67P-201 e1
ENSE00003763487	0.9	0.001	CH507-513H4.6-001 e1
ENSE00001267623	0.3	0.005	SETD2-202 e6
ENSE00001635177	0.4	0.015	MAN1A2-201 e2
ENSE00003760261	−0.4	0.015	RP11-144F15.1-001 e2
ENSE00003762709	0.7	0.015	CH507-513H4.4-001 e1
ENSE00000830378	−2.4	0.015	CAPN6-201 e2
ENSE00002089387	0.5	0.091	Y_RNA.718-201 e1
ENSE00001616358	−0.5	0.099	MST1P2-201 e11
ENSE00003903060	−0.5	0.099	GDAP1-227 e2
ENSE00003188750	−0.8	0.099	OPN1MW3-201 e6
ENSE00001544494	−0.5	0.099	MT-TQ-201 e1

**Table 6 biomolecules-15-00440-t006:** Differentially expressed exons and their transcripts in healthy controls three years after the start of the study. Nine exons (e) had a False Discovery Rate (FDR) below 0.1.

Exon ID	Log_2_ FC	FDR	Transcript and Exon
ENSE00003719556	4.7	8.2 × 10^−49^	RP11-403I13.4-002 e5
ENSE00001331364	2.3	1.9 × 10^−14^	MST1P2-201 e2
ENSE00001838554	0.9	0.000	IGHV3-21-201 e1
ENSE00003602778	1.4	0.001	RNVU1-31-201 e1
ENSE00001808682	1.5	0.010	RNVU1-2-201 e1
ENSE00003741241	0.9	0.028	RNVU1-2A-202 e1
ENSE00001807294	1.1	0.028	RNY1-201 e1
ENSE00001806816	1.3	0.032	RNU1-67P-201 e1
ENSE00001808752	0.5	0.039	SNORA13-201 e1

## Data Availability

Raw data are available from the PPMI website (www.ppmi-info.org/data, accessed on 19 January 2021).

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
