# Peer review of "The Exon-Based Transcriptomic Analysis of Parkinson’s Disease"

_biomolecules, 2025, doi:10.3390/biom15030440_

Round 1

Reviewer 1 Report (Previous Reviewer 3)

Comments and Suggestions for Authors

First, I appreciate the authors' efforts to update the introduction and results section by highlighting genes known to be related to neurodegeneration.

However, the rationale for applying a less stringent threshold in this manuscript compared to the previously published study (PMID:35289213) remains unclear without additional supporting data. I agree with the conclusion in the aforementioned paper that highly specific changes were exclusively observed in intronic transcription within the same datasets from the same cohort.

I recommend providing additional validation data, including mass spectrometry or equivalent analytical tools, to further support the major claims.

Author Response

First, I appreciate the authors' efforts to update the introduction and results section by highlighting genes known to be related to neurodegeneration.

However, the rationale for applying a less stringent threshold in this manuscript compared to the previously published study (PMID:35289213) remains unclear without additional supporting data. I agree with the conclusion in the aforementioned paper that highly specific changes were exclusively observed in intronic transcription within the same datasets from the same cohort.

As an author of the manuscript, I wanted to express my appreciation for your time and effort and for your constructive comments.

My responses are underlined and in italics.

Response: The results in our study show both FDR levels, 0.1 and 0.05. Using a threshold of 0.1 reports the genes that pass the 0.1 threshold but also gives a good opportunity to see genes with 0.05. This way, we get more transparency, and readers can see results with both stringencies when we report both threshold levels. Accordingly:

Table 3 has 2 exons with FDR 0.1 and one exon with FDR 0.05.

Table 4 has 28 exons with FDR 0.1 and 17 exons with FDR 0.05.

Table 5 has 8 exons with FDR 0.1 and 13 exons with FDR 0.05.

Table 6 has 9 exons with FDR 0.1 and 9 exons with FDR 0.05.

I think that this way, the data are represented transparently and give the readers more opportunities to understand the data. Moreover, FDR 0.1 is considered a standard FDR threshold for the Deseq2 package “Love, M.I., Huber, W. & Anders, S. Moderated estimation of fold change and dispersion for RNA-seq data with DESeq2. Genome Biol 15, 550 (2014).”

I recommend providing additional validation data, including mass spectrometry or equivalent analytical tools, to further support the major claims.

Response: We agree with the reviewer that these results must be validated. Validation is a complex process that includes literature analysis, independent sample analysis, and verification with independent analytical tools. We have provided a literature analysis because this is the only opinion we have at the moment, and we have found a perfect overlap with existing data from different cohorts. This is an independent validation.

We do not currently have additional biosamples that can be used for independent laboratory analysis with mass spectrometry or equivalent tools. Our study is exploratory in nature and only suggests using these findings for biomarker development.

I very much appreciate the reviewer’s effort and constructive comments. I hope that I adequately responded to all the comments.

Reviewer 2 Report (Previous Reviewer 2)

Comments and Suggestions for Authors

The author has made appropriate corrections to the issues pointed out. This manuscript is suitable for the publication.

Author Response

Thank you for the positive comments.

Reviewer 3 Report (Previous Reviewer 1)

Comments and Suggestions for Authors

The author replies to the questions well. I recommend to accept the article.

Author Response

Thank you for the positive comments.

Round 2

Reviewer 1 Report (Previous Reviewer 3)

Comments and Suggestions for Authors

I appreciate the author's efforts in addressing each comment. The revised manuscript has been carefully reviewed, and the provided reference supports the rationale for applying the designated threshold for analysis. Based on these improvements, I recommend this manuscript for consideration in Biomolecules.  

This manuscript is a resubmission of an earlier submission. The following is a list of the peer review reports and author responses from that submission.

Round 1

Reviewer 1 Report

Comments and Suggestions for Authors

This paper explores exon-level transcriptome analysis in Parkinson's disease (PD) to search for potential RNA biomarkers in blood. Based on the PPMI (Parkinson’s Progression Markers Initiative) dataset, the paper compared the differences in exon expression between PD patients and healthy controls at the time of diagnosis and three years after diagnosis. The results showed that despite limited changes in the blood transcriptome at diagnosis, 27 exons were significantly altered in PD patients three years later, some of which were genes related to the pathological mechanisms of PD (such as immune response and lysosomal activity). In particular, decreased expression of exons of the OPN1MW3 gene may be associated with color vision deficits in PD patients, suggesting that color vision analysis may be used as a biomarker for PD progression.

However, there are a few points worth considering.

1. Central nervous system lesions are the core feature of PD, and molecular markers in the blood may not accurately reflect the pathology in the brain. Without reliable central-peripheral correlations, the discovered biomarkers are of limited value in clinical practice.

2. The clinical application of biomarkers requires rigorous mechanism validation, which increases credibility and provides clues for new therapies. Insufficient verification weakens the persuasiveness of the article.

3. More frequent data collection will provide a more comprehensive perspective for understanding the course of PD, revealing key turning points and potential intervention windows.

If the above questions can be answered clearly, I think it will greatly enhance the article's importance and significance.

Reviewer 2 Report

Comments and Suggestions for Authors

The author reported on a search for exon gene characteristic of PD using the exon-based transcriptomic analysis. However, it’s unclear what this manuscript is trying to assert overall.

Reviewer 3 Report

Comments and Suggestions for Authors

The author explored blood-based transcriptome analysis of the PPMI cohort revealing 27 differentially expressed exons in PD patients three years after diagnosis and 13 exons changing during disease progression, primarily linked to immune response and lysosomal activity. Additionally, a decline in the OPN1MW3 gene, associated with color vision, suggests that color vision analysis could be a potential biomarker for monitoring PD progression.

This manuscript is based on the same RNA-seq datasets from the same cohorts that were previously published in Experimental Biology and Medicine in 2022 (PMID: 35289213). In the aforementioned paper, the authors claimed that highly specific changes were observed exclusively in intronic transcription, and the differentially expressed introns identified by the authors are reasonable enough to support their conclusions.  In the aforementioned paper, the author applied a significance threshold of FDR < 0.05, resulting in the following numbers of differentially expressed exons in the BL, V08, PD, and CO groups: 1, 17, 8, and 9, respectively. In this manuscript, the author used a more lenient threshold of FDR < 0.1, which increased the numbers to 2, 27, 13, and 9 for the BL, V08, PD, and CO groups, respectively. 

While the effort to identify more potential leads is acknowledgeable, applying a less stringent threshold increases the likelihood of false positives among the significant results. To strengthen the findings, the author should provide additional supporting data, such as validation through mass spectrometry or equivalent analysis tools. Once this issue is addressed, I would recommend the manuscript for consideration for publication in Biomolecules.